# Novel Approaches for the Solid-Phase Synthesis of Dihydroquinazoline-2(1*H*)-One Derivatives and Biological Evaluation as Potential Anticancer Agents

**DOI:** 10.3390/molecules27238577

**Published:** 2022-12-05

**Authors:** Qiong Wang, Ying Pan, Hongjun Luo, Yanmei Zhang, Fenfei Gao, Jinzhi Wang, Jinhong Zheng

**Affiliations:** 1Department of Chemistry, Shantou University Medical College, 22 Xinling Road, Shantou 515041, China; 2National Cancer Center/National Clinical Research Center for Cancer/Cancer Hospital & Shenzhen Hospital, Chinese Academy of Medical Sciences and Peking Union Medical College, Shenzhen 518116, China; 3Bio-Analytical Laboratory, Shantou University Medical College, 22 Xinling Road, Shantou 515041, China; 4Department of Pharmacology, Shantou University Medical College, 22 Xinling Road, Shantou 515041, China

**Keywords:** solid-phase synthesis, dihydroquinazoline-2(1*H*)-one, anticancer, antioxidation, ADMET, bioinformatics

## Abstract

In the design of antineoplastic drugs, quinazolinone derivatives are often used as small molecule inhibitors for kinases or receptor kinases, such as the EGFR tyrosine kinase inhibitor gefitinib, p38MAP kinase inhibitor **DQO-501**, and BRD4 protein inhibitor **PFI-1**. A novel and convenient approach for the solid-phase synthesis of dihydroquinazoline-2(1*H*)-one derivatives was proposed and 19 different compounds were synthesized. Cytotoxicity tests showed that most of the target compounds had anti-proliferative activity against HepG-2, A2780 and MDA-MB-231 cell lines. Among them, compounds **CA1-e** and **CA1-g** had the most potent effect on A2780 cells, with IC_50_ values of 22.76 and 22.94 μM, respectively. In addition, in an antioxidant assay, the IC_50_ of CA1-7 was 57.99 μM. According to bioinformatics prediction, ERBB2, SRC, TNF receptor, and AKT1 were predicted to be the key targets and play an essential role in cancer treatment. ADMET prediction suggested 14 of the 19 compounds had good pharmacological properties, i.e., these compounds displayed clinical potential. The correct structure of the final compounds was confirmed based on LC/MS, ^1^H NMR, and ^13^C NMR.

## 1. Introduction

Cancer poses a severe threat to human life and health and is one of the leading causes of death worldwide [1]. As a powerful means of cancer treatment, chemotherapeutic drugs have been given increasing attention [2], among which heterocyclic compounds, such as nitroimidazoles [3], thiazoles [4], and indoles [5], have been widely studied because of their special structure. Quinazolinones are nitrogen-containing heterocyclic compounds found in various alkaloids and organic molecules with bio-pharmacological activities, such as antitumor [6,7], antioxidative [8], and anti-inflammatory [9] activities. In particular, benzopyrimidinone, the core skeleton, is often used as an essential structural unit in the design of antineoplastic kinase or receptor small molecule inhibitors, such as the epidermal growth factor receptor (EGFR) tyrosine kinase inhibitor gefitinib [10], p38 mitogen-activated protein (p38MAP) kinase inhibitor DQO-501 [11,12], and bromodomain-containing protein 4 (BRD4) protein inhibitor PFI-1 (Figure 1) [6,13]. However, their success in the clinic has been limited due to the development of drug resistance, poor lipophilicity, weak selectivity, and low bioavailability [14,15,16,17]. 

Furthermore, quinazolinones have been widely used in pharmaceutical chemistry because of their heterocycles containing N-H and C-O groups, and some of their derivatives have been developed as antioxidants and potential DNA protective agents. The occurrence of cancer is caused by many factors, including free radicals, especially reactive oxygen species (ROS). Free radicals are involved in the initiation and promotion of cancer and can attack the cell membrane, cause lipid peroxidation, destroy key parts of the cell, lead to DNA strand breakage or cross-linking, change DNA molecular structure, cause gene mutation, and lead to cancer. Scavenging reactive oxygen free radicals can prevent cancer and inhibit the proliferation of malignant tumors [18]. Several reports have described solid-phase syntheses of different quinazoline and quinazoline-4-one derivatives unsatisfactorily [19,20,21,22]. Solid-phase synthesis of quinazoline-2-ones has lacked adequate investigation, and a lack of knowledge remains.

The design and synthesis of multi-functional organic small molecule libraries on solid-phase carriers has become the core of combinatorial chemistry [23]. Among small molecules, heterocyclic structures have received particular attention in combinatorial synthesis [24,25,26]. In addition, due to the long cycle of drug research and development, combinatorial chemistry high-throughput screening technology requires quinazolinones to have simple, accurate, and efficient solid-phase synthesis methods to speed up the discovery and structure optimization of lead compounds to shorten the process of new drug discovery. 3-Amino-3-(2-fluoro-5-nitrophenyl)propionic acid has been widely used as a precursor for the construction of biologically interesting heterocyclic tetrafunctional scaffolds on a solid support [27,28]. In order to exploit a wide range of applications of this precursor in solid-phase syntheses, and to maximize the chemical diversity of combinatorial libraries generated therefrom, we introduce this precursor to synthesize a series of dihydroquinazoline-2(1*H*)-one derivatives **CA1**(**a**~**j**)–**CA5**. Inspired by the successful synthesis of quinazoline derivatives based on 3-amino-3-(2-fluoro-5-nitrophenyl)propionic acid reported by Wang and co-workers [28] (Figure 1), we further investigated a novel approach to the synthesis of dihydroquinazolinones (Figure 2 and Figure 3).

In general, the attachment sites of resins in solid-phase synthesis (SPS) limit the possibility of diversity. As part of our ongoing efforts to synthesize small molecules and heterocyclic compounds, and to generate a combinatorial library of organic compounds, we developed a strategy for the solid-phase chemical route that has enabled the rapid and simplified synthesis of 1,4-disubstituted-3,4-dihydro-2(1*H*)-quinazolinones, and is designed to reference the anti-inflammatory compound DQO-501 reported by Merck and the anticancer structure PFI-1 reported by Pfizer (Figure 1). The structure has three modification sites but retains the cyclic urea structure that interacts with the receptor-binding capsule in the mother nucleus. The benzene ring or piperazine structure of the parent nuclear side chain in DQO-501 was replaced with an amide or sulfonamide structure to increase its structural diversity, improve its hydrophilicity, and improve oral bioavailability [6]. At the same time, the dichlorophenyl group at position 1-N of the mother nucleus was replaced by different fats or aromatic structures, and the side chain derived from benzoic acid at position 4-C was combined with different hydrophilic or hydrophobic amino acids to increase its structural specificity.

All target compounds designed and synthesized in this project were tested for in vitro proliferation inhibition and antioxidant activity. Secondly, considering the shortcomings of candidate drugs, such as excessive polarity, poor cell permeability, short half-life, or narrow treatment window, which are also the main reasons for the low success rate of new drug research and development, the compound structures designed in this study were based on the prediction of pharmacokinetic parameters made by ADMET (absorption, distribution, metabolism, excretion, and toxicity of drugs). This paper also preliminarily summarizes the structure–activity relationships of quinazoline-2-one-based small molecular compounds and predicts the anticancer mechanism of the compounds through bioinformatics, which provides theoretical and specific bases for the research and development of new anticancer drugs.

## 2. Results and Discussion

### 2.1. Chemistry

The synthetic strategy presented involves the synthesis of dihydroquinazoline-2(1*H*)-ones from reduced resin-bound amino acid amides using *N*-Alloc-3-amino-3-(2-fluoro-5-nitrophenyl)propionic acid as the common starting material, and the subsequent development of different building blocks using the tetrafunctional behavior of the fluoro, nitrophenyl, amino, and carboxyl groups (Figure 1).

In this study, a series of target compounds of dihydroquinazoline-2(1*H*)-one were designed and synthesized. The structure–activity relationship of drugs was evaluated according to the traditional Topliss molecular structure design method combined with comprehensive consideration of electrical activity, hydrophobicity, and stereo activity of the substituents. The target compounds exchanged R_1_ substituents with amino acids of different hydrophobic groups, and the hydrogen bonds of the R_2_ substituents and effects of other steric hindrances were studied with various primary amines. Secondly, according to the position simulation scanning method [29], the R_3_ site in the structure of dihydroquinazoline-2(1H)-one was investigated. The **CA1-a**~**j** compounds were obtained by systematic atom or group substitution of C-H bonds, linkage of an N, C-F, or C-Me to an aromatic ring or heterocyclic rings, or substitution with sulfonamide (Figure 2). The molecular, physical and chemical properties, and intramolecular and intermolecular interactions of lead compounds were varied as much as possible, while considering the limitations of drug molecules and their physical and chemical properties on biological effects, to quickly and effectively shorten the optimization period of lead compounds. At the same time, compounds **CA6-a**~**b**, and **CA7**-a~**b** were designed to further explore the effect of the substituent length at the 1-position and 4-position of the quinazolinone ring (Figure 3).

A total of 19 unreported small molecular target compounds of quinazolinone were synthesized (Table 1). All compounds were prepared and purified by RP-HPLC and characterized by LC/MS, ^1^H NMR, and ^13^C NMR spectra. Unfortunately, **CA2** synthesis was unsuccessful after the substitution of primary amines. For other compounds, LC/MS suggested that the purity of the final products would be greatly improved when the substituted primary amines were aliphatic amines, whereas when the primary amines were aromatic amines, such as benzylamine, there was an adjacent impurity peak after the peak of the main product, which increased the difficulty of purification (Table 1). When the primary amine was 2-pyridine methylamine, there was no target ion in the LC/MS spectrum of the final product. This could be due to interference of the electron on the nucleophilic substitution of the amino group on the pyridine N or self-excision of the pyridine ring without the target product during the subsequent TFA cleavage, which is consistent with previous research [28].

In addition, upon condensing the o-fluorobenzenesulfonyl chloride of compound **CA7-b** with aromatic primary amines, the main product was not the compound we designed initially. The o-fluorobenzenesulfonyl chloride not only replaced the primary amine on the benzene ring, but also replaced the 3-position imino on the lactam ring of quinazolinone in a reaction whose mechanism needs to be further explored. In the mass spectrum of the synthesized compounds, a molecular ion peak was observed as (M^+^) for all of the compounds. The liquid phase results showed that the purity of all compounds was greater than 95% (Appendix A).

To economize the action of removing amino protective groups, the Alloc amino protective group in Alloc skeleton molecule 3, originally used in OB2C combinatorial chemical solid-phase synthesis, was changed to an Fmoc defensive group (compound **3**′). Adding a deprotective group of this structure does not require the use of expensive reagents, such as phenylsilane and palladium (Figure 2), which not only shortens the reaction time but also provides a yield of more than 90% with almost no by-products. The related ^1^H NMR results are shown in Figure 2b.

Cleavage of the final product needs to occur after the solvent is removed. When we tried to separate the resin beads from the product after DMF washing, there was no cleaved product in the eluate. Further improvement was obtained by further washing the beads with methanol (MeOH) and dichloromethane (DCM) after the DMF washes, then vacuum drying for 15 min to remove the solvent from the resin beads [30]. The tert-butyl on the side chain of the amino acid was not removed after 2 h of cleavage, which was shown in the LC/MS spectrum. Considering that a long cleavage time will introduce unnecessary by-products, the optimal cleavage time was 3 h. Under these conditions, the tert-butyl group was removed, and the crude product had high purity (Figure 3).

The experimental results show that when the semi-preparative liquid phase was used to directly purify the crude product, the freeze-dried product was a very fluffy floc that was extremely unstable in the air, which made it difficult to use for subsequent characterization and biological experiments. The crude product was salted and the pH was adjusted to 3 with TFA before more than half of the liquid phase was prepared [31,32]. In addition, 0.5% TFA was added to the liquid phase to obtain the pure product as a powder, flake, or needle solid. The amino group of compound CA1-7 is directly connected to the aromatic ring, therefore this structure can easily be oxidized to unstable nitrogen oxides. After salt formation and purification, the NMR results showed a single peak at 7.88 ppm in the ^1^H NMR spectrum, which is the characteristic peak of amino salt formation on the R_2_ side chain. There were two distinct peaks for the carbonyl C-O bond in trifluoroacetic acid ion at 159.08 and 158.86 ppm in ^13^C NMR, which indicates that the two naked amino groups in the compound were salted with TFA. For compound **CA1-e**, the ^1^H NMR (DMSO-*d6*) spectra showed multiple peaks at 2.63 ppm that overlapped with the solvent peak of DMSO-*d6*, and the ^1^H NMR (D_2_O) spectra showed that the multiple peaks contained two hydrogens, corresponding to the multiple peaks at 2.73 ppm in the ^1^H NMR (DMSO-*d6*) spectra of the skeleton molecule *N*-Alloc-3-amino-3-(2-fluoro-5-nitrophenyl)propionic acid, and were the hydrogen signals of the methylene -CH_2_ in the 3-amino-propionic acid structure of the side chain of the scaffold benzene ring (Appendix A).

### 2.2. Biological Evaluation

#### 2.2.1. In Vitro Anticancer Activity

The MTT assay for cell proliferation was used to determine the 50% inhibitory concentration (IC_50_) of dihydroquinazoline-2(1*H*)-one derivatives on human liver cancer cells (HepG-2), ovarian cancer cells (A2780), and breast cancer cells (MDA-MB-231) [33,34,35], as well as human hepatic stellate cells (LX-2) (as a normal control) [36,37,38]. The above cells were tested for cytotoxicity to identify the most promising antitumor drug among the synthesized quinazolinone derivatives. Gefitinib was used as a positive control. The IC_50_s of compounds **CA1-c**, **CA1-d**, **CA1-e**, **CA1-f**, and **CA1-g** on HepG-2 cells were less than 50 μM, and the IC_50_s of compounds **CA1-e**, **CA1-g**, **CA1-i**, and **CA5** on A2780 cells were all less than 40 μM. Although the IC_50_s of every compound on MDA-MB-231 cells were all in the 70~90 μM range, they were still lower than the IC_50_s of normal cells. Therefore, the above quinazoline-2-one derivatives inhibited the growth of all three tumor cell lines in vitro and had low toxicity to normal cells. The IC_50_ values of compounds **CA1-e** and **CA1-g** on HepG-2 cells and A2780 cells were 37.59 μM and 45.41 μM, and 22.76 μM and 24.94 μM, respectively (Table 2), with **CA1-e** having the most potent cytotoxic effect. Modification at different positions of the heteroatoms on the ring of R_3_ affects activity. In this experiment (Table 1 and Table 2), the inhibitory activity of **CA1-a** (79.96 μM: A2780), **CA1-e** (22.76 μM: A2780), and **CA1-f** (>100 μM: A2780) was significantly different due to regional isomerism of nitrogen atom. In addition, compound **CA1-g**, in which R_3_ is phenoxyacetic acid, had stronger anti-proliferative activity than the other compounds. The strength of the linkage between the R_3_ ring structure and the parent nucleus of quinazolinone was reduced. For example, the growth inhibitory activity of compound **CA1-h** was lower than that of **CA1-e**, and the growth inhibitory activity was enhanced when the carbon atom on the linkage bond was replaced by oxygen atoms. **CA1-b** activity was lower than **CA1-g**. In comparing the activity of compounds **CA1-i** and **CA5**, it is not difficult to see that the activity of R_1_ is lower than that of hydroxymethyl when the side chain of R_1_ is a tetrahydropyridine ring. There was little difference between the activity of **CA1-h** and **CA3**, so the elongation of the side chain of R_1_ and the decrease of the electronegativity of R_2_ has little effect on its activity. From the comparison of compounds **CA1-g** and **CA4**, it was found that the ring structure of R_2_ reduces the activity relative to the chain structure. Compared with **CA1**–**CA5**, the activity of **CA6**–**CA7** was significantly reduced. Deletion of R_1_ or the shortening of the carbon chain of R_2_ dramatically reduces its antiproliferative activity.

In conclusion, for quinazoline-2-one derivatives, the elimination of R_1_ and the shortening of the R_2_ carbon chain may be potential factors affecting activity. An R_1_ side chain structure is better than a ring structure, but the side chain length has little effect on activity. R_2_ partial electronegativity also has little impact on activity, but as the steric hindrance increases, the activity decreases. The type of R_3_ substituent has a significant influence on activity. When the ring structure is an aromatic ring containing a pyridine ring or electron-absorbing group, its cell proliferation inhibitory activity is more potent. The connecting bond between the ring structure and the mother nucleus of quinazolinone is between the 1–2 carbon atoms. Cell proliferation inhibitory activity is enhanced after the carbon atoms on the connecting bond are replaced by electronegative oxygen atoms.

#### 2.2.2. Antioxidant Activity Measurement with the DPPH• Assay

The quinazolinone derivatives were evaluated for their free radical scavenging activity at 320, 160, 80, 40, 20, and 10 µM concentration using the DPPH radical method [39], with ascorbic acid as the positive control (Table 3). The compounds exhibited significant scavenging activity by discoloring the DPPH free radical to a stable DPPH molecule. As shown in Table 3, among the synthesized compounds, the amino-substituted compound **CA1-7** on the benzene ring displayed the most effective antioxidative activity, with an IC_50_ value of 57.99 µM, close to that of the vitamin C (Vc) standard (22.50 µM), which may be due to its aniline structure [40]. In addition, the carboxylic acid, when partially substituted with phenylmethyl (**CA1-c**) and thiophene-2-methyl groups (**CA1-j** and **CA5**), showed good antioxidant activity. The free radical scavenging ability of **CA3**, which changed the partial polarity of R_1_ and the steric hindrance of R_2_, was not significantly improved. At the same time, for **CA1-i** and **CA4**, the activity decreased when the structure of other parts remained the same, but the steric hindrance of R_1_ increased. It is speculated that the influence of the R_1_ moiety is stronger than that of the ring structure, and the type of R_3_ partial substituents will affect its antioxidant activity. Among the above compounds, R_3_ had more potent activity when it was a ring structure with strong aromaticity when the phenyl and thiophene-2-methyl groups were substituted.

Increased levels of ROS have been detected in several stages of tumorigenesis, from transformation to metastasis, in different cancer models [18]. In contrast, **CA1-e** and **CA1-g**, which are effective at killing cancer cells, showed no antioxidant activity. This result indicates that the cytotoxic effect of **CA1-e** and **CA1-g** is not associated with antioxidant properties, but rather is related to other factors, such as kinase inhibition, which is understandable, since carcinogenesis is a complex multi-step process consisting of initiation, promotion, and progression of abnormal uncontrolled cancer cells.

### 2.3. Computational Evaluation

#### 2.3.1. Bioinformatics Prediction

The bioinformatics results for target identification are shown in Figure 4, Figure 5, Figure 6 and Figure 7 and Table 4. The overlapping targets are shown in Figure 4. Protein–protein interaction (PPI) refers to the binding between two or more proteins. Through simple annotation of overlapping genes by GO enrichment analysis, we can roughly understand which biological functions, pathways, or cellular locations of overlapping genes are enriched. KEGG involves functional enrichment, whereby functions or pathways of multiple genes are associated to further explain the role of targets in metabolism, signal transduction, and other processes for studying the mechanism of drug action.

The following vital targets were identified through the above analysis: ERBB2 (HER2), the human epidermal growth factor receptor-2 tyrosine kinase [41], is overexpressed in about 25% of breast and ovarian cancers through gene amplification or transcriptional deregulation [42]. SRC, the human homolog of the Rous sarcoma virus oncogene, which encodes a non-receptor tyrosine kinase and acts as a link between inflammation and cancer, can promote tumor cell survival, angiogenesis, proliferation, and invasion. In addition, previous studies have shown that SRC family members Src and Hck are involved in the production of interleukin IL-6 in osteoblasts and immature macrophages [43]. The tumor necrosis factor (TNF) was initially identified as a glycoprotein and is present in the serum of mice infected with BCG. In general, TNF can activate pathways that lead to three different cell responses: cell survival and proliferation, transcription of pro-inflammatory genes, and cell death [44]. Phosphatidylinositol-3 kinase (PI3K) comprises a family of lipid kinases characterized by the ability to phosphorylate the inositol 3′-OH group in inositol phospholipids [45], and AKT1 (AKT serine/threonine kinase 1) is a human homolog of the v-Akt viral oncogene [46]. Studies have shown that the PI3K/AKT pathway plays a vital role in the survival and proliferation of cells overexpressing ERBB2 [47]. 

To further understand the mechanism of quinazoline-2-ones and potential tumor inhibition, we used network pharmacology and molecular docking methods to predict the molecular biological mechanism of quinazolinones in the treatment of cancer. Among the many common targets, ERBB2, SRC, TNF, and AKT1 are expected to be critical targets and play an essential role in cancer treatment. In addition, two biological processes were identified in the GO annotation. In the subsequent KEGG analysis, three enriched pathways were selected: hepatitis B, influenza A, and tuberculosis. It can be seen that quinazoline-2-one compounds may play a role in treating liver and ovarian cancer by controlling inflammatory factors and regulating immune and other biological processes.

The crystal structures of the pyridine analogue TAK-285 in complex with HER2 was described previously by Ishikawa et al [48]. In our investigations, we utilized the X-ray crystal structure of the HER2 protein (PDB ID: 3PP0), which has a low RMSD value and the resolution of X-ray crystallization. As indicated in Table 4, compounds **CA1-e** and **CA1-g** exhibited considerable binding affinity toward the binding pocket of the HER2 receptor (−61.4 and −71.2 kJ/mol, respectively). As shown in Figure 7, this class of compounds binds to the active pocket of the HER2 by forming a set of hydrogen-bonding interactions with the active amino acid residues together with other amino acid residues in the binding pocket. We confirmed that the N-H on the side chain of aliphatic amines of **CA1-e** and **CA1-h** hydrogen-bonded to the main chain NH of the hinge region (MET801 in HER2). In the costructure of **CA1-h** with HER2, the oxygen in the phenoxyethyl of R_3_ interacted with the main chain of ASP863 in the DFG motif. In addition, **CA1-h** showed similar interactions to that of TAK-285, which formed a water-mediated hydrogen bond network with the side chain of THR862. Together, these results reveal that the inhibitory activity of this class of compounds toward HER2 activity is associated with their ability to potentially bind to the active pocket of HER2 protein.

Interestingly, the IC_50_ values of compounds **CA1-e** and **CA1-g** in MDA-MB-231 and A2780 cells indicate that their effect patterns were different, which seems to be consistent with the bioinformatics prediction that HER2 might be a potential target. From the docking results of compound **CA1-e**, it can be seen that the pyridine ring interacts with HER2 in pi-pi, which may explain why the inhibitory activity of **CA1-f** on A2780 cells is lower than that of **CA1-e**. The electron-withdrawing effect of the adjacent amide bond in a conjugated system reduces the electron cloud density on the pyridine ring of **CA1-f**, thus affecting the binding of the compound to HER2. The moderate toxicity to HepG-2 may be due to the difference of targets of different cells. The MDA-MB-231 model is suitable for the screening of chemotherapeutic compounds due to its triple-negative breast cancer phenotype [49,50]. However, it should be noted that the endogenous HER2 expression in A2780 is low [51]. On the other hand, HER2 decreases drug sensitivity of ovarian cancer cells via inducing stem cell-like properties. Therefore, whether **CA1-e** and **CA1-g** exert anticancer effects through HER2 or not needs to be further explored in a HER2-positive model.

#### 2.3.2. ADMET Studies

According to Lipinski’s five rules (MW ≤ 500, nHBD ≤ 5, nHBA ≤ 10, AlogP ≤ 5, and RotB ≤ 10) for evaluation of drug-like properties of synthesized compounds [52], pharmacologically active compounds should not violate more than one of the rules. The physicochemical properties and coordination properties of the synthesized quinazoline-2-one derivatives were predicted by Discovery Studio 2016 software. As shown in Table 5, (1) the predicted values of all compounds except **CA1-g**, **CA1-h**, **CA1-i**, **CA1-j**, and **CA5** were in the appropriate range, therefore it is expected that most compounds may have potential clinical use. (2) As a critical factor of molecular chemical medicine, water solubility plays a vital role in drug distribution, in vivo transport, and transmembrane permeability. The logarithmic levels of water solubility of the compounds in the table were **3**–**5**, indicating that these compounds might have good water solubility. (3) Human intestinal absorption (HIA) is the decisive factor in the oral administration of drugs. The predicted intestinal absorption level of the above compounds was low, which is not conducive to oral administration. However, since most of the above compounds exist in alkali form, it may be possible to improve their solubility and absorption through salt formation. (4) CYP450 enzymes, which exist in the human liver, are important detoxifying enzymes and play an essential role in the metabolic stability of drugs. In this experiment, all compounds were predicted to be non-inhibitors of CYP450 enzymes, i.e., these compounds will likely not inhibit the activity of CYP450. (5) As far as distribution is concerned, none of the compounds showed blood-brain barrier permeability, indicating these drugs will not enter the brain to affect the nervous system. (6) The prediction of hepatotoxicity plays a positive role in reducing and eliminating drug-induced hepatopathy, and half of the above quinazolinone derivatives were predicted to not show hepatotoxicity. (7) The plasma protein binding rate refers to the ratio of free drug vs. drug bound to plasma protein. Only free drug can exert efficacy. The plasma protein binding rates of all the compounds in the table were 0, and the calculated AlogP values were lower than that for gefitinib (4.20), so these drugs may exist largely in the free form and have drug activity.

In summary, all compounds except **CA1-g**, **CA1-h**, **CA1-i**, **CA1-j**, and **CA5** are predicted to have oral activity and can thus become oral drugs. In the follow-up pharmacokinetic prediction, the water solubility of all compounds was higher than that of gefitinib, and CYP2D6 prediction showed the compounds were not inhibitors of CYP450 enzymes and mainly existed as the free form in plasma. Half of the drugs showed no hepatotoxicity.

Although compounds **CA1-d**, **CA1-e**, **CA1-i**, and **CA5** showed better biological activity than other compounds in the cytotoxicity test, they were also predicted to be compounds with potential hepatotoxicity. Compound **CA1-g** did not show predicted hepatotoxicity and violated two of the five Lipinski rules but was still within 95% of the acceptable range of known drugs [53]. **CA1-c** and **CA1-f** showed satisfactory drug-like and ADMET characteristics. In addition, the N atoms of the R_3_ partial pyridine ring showed predicted hepatotoxicity for compounds **CA1-a**, **CA1-e**, **CA1-d**, **CA1-i**, or **CA5**, which were substituted with fluorine atoms in the para or ortho position of the aromatic ring, or thiophene ring, i.e., the difference of the R_3_ partial substituents may affect the probability of hepatotoxicity. In addition, compounds **CA6**–**CA7** were predicted to be hepatotoxic compounds. The deletion of R_1_ or the shortening of the R_2_ carbon chain may be potential influencing factors of their hepatotoxicity.

## 3. Experimental Section

### 3.1. Materials and Instruments

Fmoc-protected amino acids: Fmoc-D-Ser(tBu)-OH, Fmoc-Orn(Boc)-OH, Fmoc-Aib-OH, Fmoc-Aad(OtBu)-OH, Fmoc-4-Apc(Boc)-OH. Primary amine: *N*-Boc-1,3-propanediamine, benzylamine, isopentylamine, 2-(aminomethyl)pyridine, methylamine hydrochloride. Carboxylic acids: isonicotinic acid, benzoic acid, p-methylbenzoic acid, p-hydroxybenzoic acid, p-fluorobenzoic acid, nicotinic acid, 2-pyridinecarboxylic acid, phenoxyacetic acid, 3-pyridinepropionic acid, 2-thiopheneacetic acid. Sulfonyl chlorides: o-methoxybenzenesulfonyl chloride, o-fluorobenzenesulfonyl chloride. Commercially available reagents were used without further purification unless otherwise stated. Preparation of Kaiser test reagent: Solution A: pyridine; Solution B: 80% phenol alcohol solution; Solution C: dissolve 0.1 g ninhydrin in 10 mL ethanol.

The HepG-2, A2780, MDA-MB-231, and LX-2 cell lines were obtained from Boster Biological Technology Co., Ltd. (Wuhan, China).

Solid synthesis was performed on a QB-210 rotating shaker (Kylin-Bell, Haimen, China). Melting points were determined using a WRS-1B digital melting point meter (Shenguang, Shanghai, China). Nitrogen blowing and freeze-drying were completed on a BF2000 nitrogen blow dryer (BafangShiji, Hangzhou, China) and a CoolSafe110-4 freeze dryer (LaboGene, Lynge, Denmark), respectively. Absorbance was determined by an Infinite M200 Pro raster full-wavelength enzyme labeling instrument (Tecan, Männedorf, Switzerland).

RP-HPLC was carried out on a Dalian Elite analytical instrument (Fullerton, Dalian, China). Purification of samples was performed using a SinoChrom ODS-BP column, 10 μm, 20.0 mm × 250 mm. All HPLC experiments were performed using gradient elution: solvent A (H_2_O with 0.05% TFA) and solvent B [acetonitrile (CH_3_CN) with 0.05% TFA]. Flow rates were 0.2 and 17.0 mL/min for analytical and preparative chromatograms, respectively, using a λ = 254 nm.

LC/MS (ESI) was recorded on an LCMS-2020 system (Shimadzu, Kyoto, Japan) at λ = 254 nm using an Accucore C18, 2.6 µm, 100 × 3.0 mm column.

^1^H and ^13^C NMR spectra were recorded in DMSO-*d_6_* solutions at 600 MHz (^1^H) and 151 MHz (^13^C) on a Bruker Avance NEO 600 spectrometer (Bruker, Saarbrücken, Germany) using tetramethylsilane (TMS) as the internal standard. The yield for the purified products was determined based on the loading of the polymeric support starting from 1.00 g of the resin.

### 3.2. Chemistry

#### 3.2.1. Synthesis of the Tetrafunctional Scaffold

Compound **1**–**3** was synthesized according to Wang et al. [29].

Synthesis of 3-amino-3-(2-fluorophenyl)propionic acid **1**: 2-fluorobenzaldehyde (3.00 g, 24 mmol), malonic acid (2.75 g, 26 mmol), and ammonium acetate (4.10 g, 0.53 mmol) were mixed in 25 mL of ethanol and heated under reflux overnight. The reaction mixture was cooled to room temperature, and the white solid was collected by filtration and washed with ethanol three times (3 × 50 mL) and ether two times (2 × 50 mL). Then the solid was dried in vacuo. Weight: 2.01 g, yield: 58.6%.

Synthesis of 3-amino-3-(2-fluoro-5-nitrophenyl)propionic acid **2**: The dry solid 3-amino-3-(2-fluorophenyl)propionic acid **1** was added into an ice-chilled mixture of HNO_3_/H_2_SO_4_ (8 mL/8 mL) and stirred for 6 h in an ice bath. The yellowish solution was poured into 40 mL of ice and neutralized with 1.25 mol/L sodium hydroxide (NaOH) to pH = 6 and allowed to warm to room temperature. The precipitate was collected by suction-filtration, then washed with water (25 mL), ethanol (3 × 50 mL), and ether (50 mL). Weight: 1.52 g, yield: 60.6%.

Synthesis of *N*-Alloc-3-amino-3-(2-fluoro-5-nitrophenyl)propionic acid **3**: To the solution of 3-amino-3-(2-fluoro-5-nitrophenyl)propionic acid **2** (1.00 g, 4.4 mmol) in aqueous sodium hydrogen carbonate (NaHCO_3_) solution (0.92 g, 11 mmol), 30 mL of Alloc-OSu (0.87 g, 4.4 mmol) solution in CH_3_CN (15 mL) was added dropwise. The mixture was stirred at room temperature overnight and washed with ethyl acetate twice (2 × 40 mL). The aqueous solution was acidified with muriatic acid (HCl) to pH = 3, and the resultant solidified powder was collected by filtration, washed with water (5 × 50 mL), and dried in vacuo. Weight: 0.88 g, yield: 64%.

#### 3.2.2. Typical Procedure for the Synthesis of Resin-Bound Quinazolinones

A total of 1.00 g Rink amide-MBHA resin (1% divinylbenzene, 100–200 mesh, 0.432 mmol/g substitutions) in a disposable fritted polypropylene column (20 mL) was swollen in DMF, shaken for 24 h at room temperature, and drained.

(a)Coupling of an Amino Acid to the Resin

The swollen Rink resin was deprotected with 20% 4-methylpiperidine for 5–15 min and washed with DMF (3×), followed by MeOH (3×) and DMF (3×), then coupled with Fmoc-D-Ser(tBu)-OH (3 equiv, 1.296 mmol) and 1-hydroxybenzotriazole (HOBt) (3.03 equiv, 1.309 mmol) in DMF (15 mL). DIC (3.06 equiv, 1.322 mmol) was then added and the mixture was shaken vigorously at room temperature for 5 h. After removal of the liquid by filtration, the beads were washed with DMF (3×), MeOH (3×), and DMF (3×). A negative ninhydrin test [54] confirmed the completeness of the coupling reactions.

(b)*N*-Acylation with *N*-Alloc-3-amino-3-(2-fluoro-5-nitrophenyl)propionic acid

The Fmoc was removed with 20% 4-methylpiperidine (5 min, 15 min) followed by washing, then a mixture of *N*-Alloc-3-amino-3-(2-fluoro-5-nitrophenyl) propionic acid (3 equiv, 1.296 mmol) and HOBt (3.03 equiv, 1.309 mmol) in DMF (15 mL) was added to DIC (3.06 equiv, 1.322 mmol). The reaction was shaken at room temperature for 8 h and the excess reagent was drained. The resin beads were washed, alternating with DMF (3×), MeOH (3×), and DMF (3×). Completeness of the coupling was verified by a ninhydrin test.

(c)Nucleophilic Substitution of the Fluoro Group

The resulting o-fluoro-nitro derivative was treated with a primary amine (20 equiv, 8.64 mmol) in the presence of DIEA (20 equiv, 8.64 mmol) overnight at room temperature, and the excess reagent was drained. The resin was washed with DMF (3×), MeOH (3×), and DCM (3×) before the next step.

(d)Removal of the Alloc Protective Base

The Alloc was removed with phenylsilane (PhSiH_3_) (20 equiv, 17.28 mmol) and tetrakis(triphenylphosphine)palladium(0) (Pd(PPh_3_)_4_) (0.2 equiv, 0.17 mmol) in DCM for 50 min, two times, then washed with DCM (6×), DMF (3×), 5% DIEA in DMF (3×), 5% sodium diethyldithiocarbamate trihydrate (3×), 50% DCM in DMF (3×), MeOH (3×), DMF (3×), and DCM (3×). Completion of the reaction was confirmed by a positive ninhydrin test.

(e)Cyclization using Carbonyldiimidazole

The resin-bound 2-aminomethylaniline compound was cyclized by treatment with CDI (4 equiv, 3.46 mmol) and pyridine (6 equiv, 5.18 mmol) in DCM, followed by washes with DCM (3×), MeOH (3×), and DMF (3×). Although carbonylations with carbonyldiimidazole (CDI) were slow, we decided to opt for annulation with CDI, as no significant amount of byproducts were observed compared with triphosgene or 4-nitrophenyl chloroformate.

(f)Reduction of the Aromatic Nitro Group

The resulting resin-bound 6-nitroquinazolinone was treated with tin(II) chloride dihydrate (20 equiv, 2 M) in DMF for 24 h at room temperature, followed by washes with DMF (3×), MeOH (3×), and DMF (3×) to generate the 6-aminoquinazolinone analog.

(g)Coupling with Carboxylic Acid/Sulfonyl Chloride

A solution of carboxylic acids (10 equiv, 4.32 mmol), HATU (10 equiv, 4.32 mmol), and 4-dimethylaminopyridine (DMAP) (0.15 equiv, 0.06 mmol) dissolved in anhydrous DMF was added to DIEA (20 equiv, 8.64 mmol), or a solution of sulfonyl chloride (10 equiv, 4.32 mmol) and DIEA (10 equiv, 4.32 mmol) in DCM. The solution was added to the resin and shaken at room temperature overnight. The excess reagent was drained and the resin was washed in the same manner as above. Completion of the reaction was confirmed by a negative ninhydrin test.

All washes with DMF, MeOH, DCM, 5% DIEA in DMF, or 5% sodium diethyldithiocarbamate trihydrate in DMF were ~2 min each. The beads were washed with DMF (3×), MeOH (3×), and DCM (3×), respectively, three times and then dried under a vacuum for 15 min. Side chain deprotection was achieved using 15 mL of a TFA cocktail (a mixture of 82.5% TFA; 5% phenol; 5% thioanisole; 5% water; and 2.5% triisopropylsilane (TIS), *v*/*v*) for 3 h. The solution was collected into a 50 mL polypropylene tube, the resin was washed with two portions of TFA (5 mL each), and the washes were combined and concentrated with a low stream of air until all TFA was removed. The product was precipitated using cold diethyl ether and the solid was washed twice with cold ether. The crude product was air-dried before purification by reversed-phase high-performance liquid chromatography (RP-HPLC).

*N*-Fmoc-3-amino-3-(2-fluoro-5-nitrophenyl)propionic acid **3**’: white solid; yield: 68.2%; mp 200.8–201.2 °C; ^1^H NMR (600 MHz, DMSO) δ 8.36 (dd, *J* = 6.1, 2.8 Hz, 1H), 8.22 (dd, *J* = 8.7, 3.9 Hz, 2H), 7.87 (t, *J* = 7.2 Hz, 2H), 7.65 (dd, *J* = 15.3, 7.5 Hz, 2H), 7.49 (t, *J* = 9.1 Hz, 1H), 7.39 (dt, *J* = 11.9, 7.5 Hz, 2H), 7.27 (dt, *J* = 7.2, 3.6 Hz, 2H), 5.27 (td, *J* = 8.6, 6.1 Hz, 1H), 4.34–4.28 (m, 2H), 4.20 (t, *J* = 6.8 Hz, 1H), 2.74 (ddd, *J* = 22.1, 16.2, 7.5 Hz, 2H); ^13^C NMR (151 MHz, DMSO) δ 171.60, 164.11, 162.41 (C-F, d, ^1^*J*_C-F_ = 255.86 Hz), 155.75, 144.63, 144.30, 143.98, 141.19, 141.16, 132.24, 132.13 (C-F, d, ^2^*J*_C-F_ = 16.5 Hz), 128.12, 128.06 (C-F, d, ^3^*J*_C-F_ = 8.68 Hz), 127.48, 127.45, 125.71, 125.64 (C-F, d, ^3^*J*_C-F_ = 10.61 Hz), 125.54, 125.52, 124.36, 124.32, 120.63, 120.57, 117.61, 117.44 (C-F, d, ^2^*J*_C-F_ = 24.60 Hz), 65.94, 47.09, 45.6; MS (ESI) (*m*/*z*): 450.12, [M + H^+^]: 451.10.3-{4-{2-[(1-amino-3-hydroxy-1-oxopropan-2-yl)amino]-2-oxoethyl}-6-nitro-2-oxo-3,4-dihydroquinazolin-1(2*H*)-yl}propan-1-aminium 2,2,2-trifluoroacetate (**CA1-6**)**·TFA**: yellow solid; yield 93%; ^1^H NMR (600 MHz, DMSO) δ 8.15–8.10 (m, 1H), 8.07 (dd, *J* = 25.3, 2.5 Hz, 1H), 7.96 (d, *J* = 8.0 Hz, 1H), 7.86 (s, 3H), 7.52 (dd, *J* = 53.4, 2.7 Hz, 1H), 7.34–7.19 (m, 2H), 7.08 (d, *J* = 9.3 Hz, 1H), 4.89 (dtd, *J* = 25.9, 6.5, 2.8 Hz, 1H), 4.19 (td, *J* = 10.0, 5.1 Hz, 1H), 3.97–3.90 (m, 2H), 3.56–3.39 (m, 2H), 2.90 (d, *J* = 5.8 Hz, 2H), 2.70–2.53 (m, 2H), 1.97–1.75 (m, 2H); ^13^C NMR (151 MHz, DMSO) δ 172.39 (d, *J* = 6.2 Hz), 169.27, 158.74 (C-F, q, ^2^*J*_C-F_ = 31.6 Hz), 153.15, 143.77, 141.54, 124.69, 124.22, 122.70, 117.46 (C-F, q, ^1^*J*_C-F_ = 299.3 Hz), 114.01, 62.08, 55.25, 50.07, 44.45, 37.07, 25.46; MS (ESI) (*m*/*z*): 394. 16, [M + H^+^]: 395.15.4-{2-[(1-amino-3-hydroxy-1-oxopropan-2-yl)amino]-2-oxoethyl}-1-(3-ammoniopropyl)-2-oxo-1,2,3,4-tetrahydroquinazolin-6-aminium 2,2,2-trifluoroacetate (**CA1-7**)**·TFA**: brown solid; yield 76%; ^1^H NMR (600 MHz, DMSO) δ 7.99 (dd, *J* = 20.8, 7.9 Hz, 1H), 7.88 (s, 1H), 7.33 (d, *J* = 14.9 Hz, 1H), 7.13 (ddd, *J* = 30.6, 18.0, 5.7 Hz, 2H), 7.05 (d, *J* = 1.8 Hz, 1H), 4.76 (dtd, *J* = 19.5, 6.6, 2.7 Hz, 1H), 4.23 (dd, *J* = 12.3, 5.5 Hz, 1H), 3.93–3.74 (m, 1H), 3.56 (dddd, *J* = 31.8, 26.5, 10.2, 5.4 Hz, 1H), 2.94–2.83 (m, 1H), 2.55 (td, *J* = 14.8, 7.5 Hz, 1H), 1.93–1.75 (m, 1H); ^13^C NMR (151 MHz, DMSO) δ 172.52, 169.60, 159.08, 158.86 (C-F, q, ^2^*J*_C-F_ = 32.4 Hz), 154.02, 135.78, 125.03, 121.84, 120.15, 119.92, 118.23, 116.25 (C-F, q, ^1^*J*_C-F_ = 299.3 Hz), 114.55, 62.02, 55.46, 50.23, 44.42, 39.05, 37.13, 25.48; MS (ESI) (*m*/*z*): 364.19, [M + H^+^]: 365.20.3-{4-{2-[(1-amino-3-hydroxy-1-oxopropan-2-yl)amino]-2-oxoethyl}-6-(isonicotinamido)-2-oxo-3,4-dihydroquinazolin-1(2*H*)-yl}propan-1-aminium 2,2,2-trifluoroacetate (**CA1-a**)**·TFA**: orange solid; yield 86%; ^1^H NMR (600 MHz, DMSO) δ 10.53 (d, *J* = 19.5 Hz, 1H), 8.85 (s, 2H), 8.06–7.93 (m, 3H), 7.79 (s, 3H), 7.68 (ddd, *J* = 18.6, 8.8, 2.3 Hz, 1H), 7.62 (s, 1H), 7.32 (d, *J* = 14.5 Hz, 1H), 7.18–7.11 (m, 1H), 7.11–7.00 (m, 2H), 4.80 (dd, *J* = 9.3, 3.8 Hz, 1H), 4.26 (dt, *J* = 7.8, 5.4 Hz, 1H), 3.94–3.84 (m, 2H), 3.54 (ddd, *J* = 15.9, 10.9, 5.5 Hz, 2H), 2.89 (dd, *J* = 12.5, 6.2 Hz, 2H), 2.73–2.52 (m, 2H), 1.88 (ddt, *J* = 19.1, 13.6, 6.7 Hz, 2H); ^13^C NMR (151 MHz, DMSO) δ 172.58, 169.80, 163.55, 158.94 (C-F, q, ^2^*J*_C-F_ = 32.4 Hz), 154.15, 149.58, 143.49, 134.05, 133.18, 124.04, 122.58, 120.97, 119.13 (C-F, q, ^1^*J*_C-F_ = 247.39 Hz), 113.79, 62.07, 55.54, 50.69, 44.73, 38.89, 37.19, 25.65; MS (ESI) (*m*/*z*): 469.21, [M + H^+^]: 470.15.3-{4-{2-[(1-amino-3-hydroxy-1-oxopropan-2-yl)amino]-2-oxoethyl}-6-benzamido-2-oxo-3,4-dihydroquinazolin-1(2*H*)-yl}propan-1-aminium 2,2,2-trifluoroacetate (**CA1-b**)**·TFA**: yellow solid; yield 85%; ^1^H NMR (600 MHz, DMSO) δ 10.19 (d, *J* = 12.5 Hz, 1H), 8.04–7.97 (m, 1H), 7.97–7.92 (m, 2H), 7.74 (s, 3H), 7.70–7.65 (m, 1H), 7.65–7.62 (m, 1H), 7.60 (t, *J* = 7.3 Hz, 1H), 7.54 (td, *J* = 7.6, 1.7 Hz, 2H), 7.32 (d, *J* = 21.9 Hz, 1H), 7.10 (d, *J* = 10.7 Hz, 1H), 7.08–6.96 (m, 2H), 4.81–4.73 (m, 1H), 4.26 (dq, *J* = 7.7, 5.4 Hz, 1H), 3.88 (dtt, *J* = 21.2, 14.3, 7.0 Hz, 2H), 2.89 (dd, *J* = 12.6, 6.5 Hz, 2H), 2.65–2.52 (m, 2H), 1.94–1.82 (m, 2H); ^13^C NMR (151 MHz, DMSO) δ 172.52, 169.79, 165.64, 158.62, 158.39 (C-F, q, ^2^*J*_C-F_ = 34.82 Hz), 154.18, 135.25, 133.90, 133.51, 132.00, 128.88, 128.02, 123.94, 120.83, 119.01, 113.69 (C-F, q, ^1^*J*_C-F_ = 275.78 Hz), 62.11, 55.48, 50.67, 44.71, 38.87, 37.22, 25.67; MS (ESI) (*m*/*z*): 468.21, [M + H^+^]: 469.20.3-{4-{2-[(1-amino-3-hydroxy-1-oxopropan-2-yl)amino]-2-oxoethyl}-6-(4-methylbenzamido)-2-oxo-3,4-dihydroquinazolin-1(2*H*)-yl}propan-1-aminium 2,2,2-trifluoroacetate (**CA1-c**)**·TFA**: white solid; yield 82%; ^1^H NMR (600 MHz, DMSO) δ 10.10 (d, *J* = 11.8 Hz, 1H), 7.99 (dd, *J* = 40.7, 7.9 Hz, 1H), 7.88 (dd, *J* = 10.2, 8.2 Hz, 2H), 7.73 (s, 3H), 7.67 (ddd, *J* = 14.4, 8.8, 2.1 Hz, 1H), 7.62 (s, 1H), 7.37–7.28 (m, 3H), 7.10 (d, *J* = 9.6 Hz, 1H), 7.07–6.95 (m, 2H), 4.81–4.72 (m, 1H), 4.26 (dq, *J* = 7.7, 5.4 Hz, 1H), 3.88 (pd, *J* = 14.2, 6.7 Hz, 2H), 3.59 (dd, *J* = 10.6, 5.5 Hz, 2H), 2.88 (dd, *J* = 12.6, 6.3 Hz, 2H), 2.63–2.52 (m, 2H), 2.39 (s, 3H), 1.93–1.83 (m, 2H); ^13^C NMR (151 MHz, DMSO) δ 172.56, 169.88, 169.73 (C-F, q, ^2^*J*_C-F_ = 22.39 Hz), 165.43, 154.17, 142.01, 133.98, 133.41, 132.40, 132.34, 129.40, 128.05, 120.81, 119.00 (C-F, q, ^1^*J*_C-F_ = 279.55 Hz), 118.32, 113.66, 62.09, 55.49, 50.67, 44.70, 38.82, 37.22, 25.67, 21.47; MS (ESI) (*m*/*z*): 482.23, [M + H^+^]: 483.20.3-{4-{2-[(1-amino-3-hydroxy-1-oxopropan-2-yl)amino]-2-oxoethyl}-6-(4-fluorobenzamido)-2-oxo-3,4-dihydroquinazolin-1(2*H*)-yl}propan-1-aminium 2,2,2-trifluoroacetate (**CA1-d**)**·TFA**: yellow solid; yield 79%; ^1^H NMR (600 MHz, DMSO) δ 10.20 (d, *J* = 16.2 Hz, 1H), 8.11–7.89 (m, 3H), 7.73 (s, 3H), 7.66 (ddd, *J* = 17.0, 8.9, 2.2 Hz, 1H), 7.61 (s, 1H), 7.43–7.35 (m, 2H), 7.32 (d, *J* = 25.3 Hz, 1H), 7.10 (d, *J* = 13.3 Hz, 1H), 7.08–6.96 (m, 2H), 4.83–4.71 (m, 1H), 4.29–4.22 (m, 1H), 3.95–3.83 (m, 2H), 2.89 (dd, *J* = 12.5, 6.3 Hz, 2H), 2.64–2.52 (m, 2H), 1.88 (ddd, *J* = 21.5, 10.8, 5.5 Hz, 2H); ^13^C NMR (151 MHz, DMSO) δ 172.57, 169.77, 164.55, 163.69 (C-F, q, ^1^*J*_C-F_ = 249.32 Hz), 158.38 (C-F, q, ^2^*J*_C-F_ = 31.83 Hz), 154.15, 133.78, 133.56, 131.69, 130.75, 123.95, 120.83, 119.02 (C-F, q, ^1^*J*_C-F_ = 274.60 Hz), 115.88, 115.74, 113.70, 62.10, 55.49, 50.66, 44.65, 38.84, 37.19, 25.66; MS (ESI) (*m*/*z*): 486.20, [M + H^+^]: 487.20.3-{4-{2-[(1-amino-3-hydroxy-1-oxopropan-2-yl)amino]-2-oxoethyl}-6-(nicotinamido)-2-oxo-3,4-dihydroquinazolin-1(2*H*)-yl}propan-1-aminium 2,2,2-trifluoroacetate (**CA1-e**)**·TFA**: yellow solid; yield 84%; ^1^H NMR (600 MHz, DMSO) δ 10.45 (d, *J* = 15.6 Hz, 1H), 9.15 (d, *J* = 8.4 Hz, 1H), 8.80 (d, *J* = 4.6 Hz, 1H), 8.38 (ddd, *J* = 9.7, 8.0, 4.0 Hz, 1H), 7.99 (dd, *J* = 39.8, 7.9 Hz, 1H), 7.79 (s, 3H), 7.67 (dd, *J* = 11.7, 3.1 Hz, 1H), 7.65–7.63 (m, 1H), 7.62 (dd, *J* = 6.4, 2.2 Hz, 1H), 7.31 (s, 1H), 7.10 (dd, *J* = 12.0, 9.7 Hz, 2H), 7.07–6.99 (m, 1H), 4.80–4.74 (m, 1H), 4.27–4.25 (m, 1H), 3.93–3.82 (m, 2H), 3.62–3.48 (m, 2H), 2.89 (dd, *J* = 12.6, 6.3 Hz, 2H), 2.64–2.52 (m, 2H), 1.94–1.83 (m, 2H); ^13^C NMR (151 MHz, DMSO) δ 172.58, 169.75, 163.82, 158.67 (C-F, q, ^2^*J*_C-F_ = 35.9 Hz), 154.17, 151.69, 148.35, 136.83, 133.83, 133.47, 131.23, 124.40, 124.02, 120.84, 119.02, 117.32, 115.38 (C-F, q, ^1^*J*_C-F_ = 283.7 Hz), 113.77, 62.06, 55.47, 50.65, 44.65, 38.88, 37.20, 25.66; MS (ESI) (*m*/*z*): 469.21, [M + H^+^]: 470.20.3-{4-{2-[(1-amino-3-hydroxy-1-oxopropan-2-yl)amino]-2-oxoethyl}-2-oxo-6-(picolinamido)-3,4-dihydroquinazolin-1(2*H*)-yl}propan-1-aminium 2,2,2-trifluoroacetate (**CA1-f**)**·TFA**: orange solid; yield 76%; ^1^H NMR (600 MHz, DMSO) δ 10.67–10.53 (m, 1H), 8.79–8.69 (m, 1H), 8.15 (d, *J* = 7.8 Hz, 1H), 8.07 (t, *J* = 7.7 Hz, 1H), 7.99 (dd, *J* = 17.4, 7.9 Hz, 1H), 7.78 (dd, *J* = 7.1, 4.4 Hz, 5H), 7.68 (dd, *J* = 6.6, 4.9 Hz, 1H), 7.29 (s, 1H), 7.08 (d, *J* = 7.3 Hz, 1H), 7.01 (dd, *J* = 52.7, 2.0 Hz, 2H), 4.84–4.74 (m, 1H), 4.33–4.20 (m, 1H), 3.97–3.80 (m, 2H), 3.65–3.49 (m, 2H), 2.89 (dt, *J* = 12.7, 6.3 Hz, 2H), 2.65–2.51 (m, 2H), 1.97–1.78 (m, 2H); ^13^C NMR (151 MHz, DMSO) δ 172.54, 169.82, 162.62, 158.88 (C-F, q, ^2^*J*_C-F_ = 34.93 Hz), 154.11, 150.32, 148.88, 138.61, 133.77, 133.07, 127.33, 123.94, 122.71, 120.74, 118.90, 117.17 (C-F, q, ^1^*J*_C-F_ = 71.71 Hz), 113.78, 62.08, 55.48, 50.68 (d, *J* = 12.5 Hz), 44.58, 38.88, 37.20, 25.66; MS (ESI) (*m*/*z*): 469.21, [M + H^+^]: 470.15.3-{4-{2-[(1-amino-3-hydroxy-1-oxopropan-2-yl)amino]-2-oxoethyl}-2-oxo-6-(2-phenoxyacetamido)-3,4-dihydroquinazolin-1(2*H*)-yl}propan-1-aminium 2,2,2-trifluoroacetate (**CA1-g**)**·TFA**: white solid; yield 85%; ^1^H NMR (600 MHz, DMSO) δ 10.03 (d, *J* = 14.1 Hz, 1H), 7.98 (dd, *J* = 37.5, 7.9 Hz, 1H), 7.77 (s, 3H), 7.58–7.51 (m, 1H), 7.4 (dd, *J* = 14.2, 2.0 Hz, 1H), 7.32 (t, *J* = 7.8 Hz, 2H), 7.30 (d, *J* = 3.8 Hz, 1H), 7.10 (s,1H), 7.07–6.95 (m, 5H), 4.87 (d, *J* = 34.2 Hz, 1H), 4.78–4.69 (m, 1H), 4.60 (d, *J* = 4.2 Hz, 2H), 4.24 (dq, *J* = 10.4, 5.2 Hz, 1H), 3.93–3.79 (m, 2H), 3.55 (d, *J* = 42.7 Hz, 2H), 2.87 (d, *J* = 5.4 Hz, 2H), 2.62–2.51 (m, 2H), 1.86 (dq, *J* = 12.8, 6.9 Hz, 2H); ^13^C NMR (151 MHz, DMSO) δ 172.63, 172.54, 169.73, 163.55, 158.83 (C-F, q, ^2^*J*_C-F_ = 30.59 Hz), 154.12, 149.83, 149.58, 143.49, 134.05, 133.18, 124.04, 122.58, 120.97, 119.13, 115.48 (C-F, q, ^1^*J*_C-F_ = 255.65 Hz), 113.79, 62.07, 55.48, 51.80, 50.63, 44.73, 38.92, 37.19, 25.65; MS (ESI) (*m*/*z*): 498.22, [M + H^+^]: 499.20.3-{4-{2-[(1-amino-3-hydroxy-1-oxopropan-2-yl)amino]-2-oxoethyl}-2-oxo-6-[3-(pyridin-3-yl)propanamido]-3,4-dihydroquinazolin-1(2*H*)-yl}propan-1-aminium 2,2,2-trifluoroacetate (**CA1-h**)**·TFA**: yellow solid; yield 92%; ^1^H NMR (600 MHz, DMSO) δ 9.96 (d, *J* = 15.4 Hz, 1H), 8.78 (s, 1H), 8.70 (s, 1H), 8.27 (d, *J* = 7.6 Hz, 1H), 7.98 (dd, *J* = 38.9, 7.9 Hz, 1H), 7.85 (s, 3H), 7.83–7.80 (m, 1H), 7.42 (s, 1H), 7.40 (dd, *J* = 5.0, 2.6 Hz, 1H), 7.30 (d, *J* = 6.5 Hz, 1H), 7.11 (d, *J* = 7.3 Hz, 1H), 7.06–6.92 (m, 2H), 4.83–4.65 (m, 1H), 4.24 (dtd, *J* = 7.8, 5.3, 2.4 Hz, 1H), 3.89–3.79 (m, 2H), 3.64–3.44 (m, 2H), 3.06 (t, *J* = 7.1 Hz, 2H), 2.87 (dd, *J* = 12.3, 6.2 Hz, 2H), 2.74 (d, *J* = 7.2 Hz, 2H), 2.65–2.51 (m, 2H), 1.93–1.78 (m, 2H); ^13^C NMR (151 MHz, DMSO) δ 172.55, 169.86, 169.68, 159.02, 158.80 (C-F, q, ^2^*J*_C-F_ = 33.5 Hz), 154.18, 144.44, 143.33, 142.52, 140.36, 133.79, 133.09, 130.88, 126.24, 124.05, 119.56, 118.02, 117.71, 116.06 (C-F, q, ^1^*J*_C-F_ = 294.9 Hz), 113.78, 62.06, 55.47, 50.58, 44.56, 38.84, 37.14, 36.86, 28.04, 25.63; MS (ESI) (*m*/*z*): 497.24, [M + H^+^]: 498.25.3-{4-{2-[(1-amino-3-hydroxy-1-oxopropan-2-yl)amino]-2-oxoethyl}-2-oxo-6-[2-(thiophen-2-yl)acetamido]-3,4-dihydroquinazolin-1(2*H*)-yl}propan-1-aminium 2,2,2-trifluoroacetate (**CA1-i**)**·TFA**: pale brown solid; yield 90%; ^1^H NMR (600 MHz, DMSO) δ 10.18 (d, *J* = 15.2 Hz, 1H), 7.99 (dd, *J* = 38.4, 7.9 Hz, 1H), 7.8 (s, 3H), 7.45 (dd, *J* = 23.6, 5.1 Hz, 2H), 7.41–7.35 (m, 1H), 7.30 (d, *J* = 4.2 Hz, 1H), 7.10 (d, *J* = 9.6 Hz, 1H), 7.08–6.90 (m, 4H), 4.77–4.67 (m, 1H), 4.28–4.21 (m, 1H), 3.86 (t, *J* = 9.1 Hz, 4H), 3.61–3.48 (m, 2H), 2.93–2.82 (m, 2H), 2.62–2.51 (m, 2H), 1.86 (ddd, *J* = 22.5, 10.5, 4.8 Hz, 2H); ^13^C NMR (151 MHz, DMSO) δ 172.56, 169.79, 168.19, 158.70 (C-F, q, ^2^*J*_C-F_ = 32.3 Hz), 154.15, 137.64, 133.80, 133.27, 129.82, 127.11, 126.77, 125.49, 124.12, 119.62, 117.82, 115.69 (C-F, q, ^1^*J*_C-F_ = 321.4 Hz), 113.82, 62.06, 55.46, 50.57, 44.53, 38.83, 37.86, 37.16, 25.63; MS (ESI) (*m*/*z*): 488.18, [M + H^+^]: 489.20.3-{4-{2-[(1-amino-3-hydroxy-1-oxopropan-2-yl)amino]-2-oxoethyl}-6-[(2-methoxyphenyl)sulfonamido]-2-oxo-3,4-dihydroquinazolin-1(2*H*)-yl}propan-1-aminium 2,2,2-trifluoroacetate (**CA1-j**)**·TFA**: yellow solid; yield 71%; ^1^H NMR (600 MHz, DMSO) δ 9.73 (d, *J* = 35.7 Hz, 1H), 7.96 (dd, *J* = 10.1, 8.1 Hz, 1H), 7.75 (s, 3H), 7.72–7.68 (m, 1H), 7.59–7.54 (m, 1H), 7.33 (d, *J* = 24.0 Hz, 1H), 7.18 (d, *J* = 8.4 Hz, 1H), 7.12 (d, *J* = 22.3 Hz, 1H), 7.02 (td, *J* = 7.5, 2.7 Hz, 1H), 6.99–6.90 (m, 3H), 6.87 (d, *J* = 8.1 Hz, 1H), 4.67–4.59 (m, 1H), 4.24 (dt, *J* = 7.7, 5.3 Hz, 1H), 3.90 (s, 3H), 3.81–3.71 (m, 2H), 3.63–3.53 (m, 2H), 2.82 (d, *J* = 6.0 Hz, 2H), 2.41 (tt, *J* = 14.6, 7.2 Hz, 2H), 1.78 (ddd, *J* = 19.9, 11.7, 4.8 Hz, 2H); ^13^C NMR (151 MHz, DMSO) δ 172.53, 169.57, 158.81 (C-F, q, ^2^*J*_C-F_ = 33.8 Hz), 154.15, 153.99, 135.47, 134.17, 132.17, 130.62, 126.85, 124.46, 120.98, 120.55, 119.43, 119.62, 116.85 (C-F, q, ^1^*J*_C-F_ = 294.6 Hz), 113.24, 62.11, 56.56, 55.45, 50.29, 44.42, 38.88, 37.14, 25.53; MS (ESI) (*m*/*z*): 534.19, [M + H^+^]: 535.20.5-amino-4-{2-{1-isopentyl-2-oxo-6-[3-(pyridin-3-yl)propanamido]-1,2,3,4-tetrahydroquinazolin-4-yl}acetamido}-5-oxopentan-1-aminium 2,2,2-trifluoroacetate (**CA3**)**·TFA**: light brown solid; yield 74%; ^1^H NMR (600 MHz, DMSO) δ 9.95 (d, *J* = 18.9 Hz, 1H), 8.73 (d, *J* = 45.9 Hz, 1H), 8.25 (dd, *J* = 12.5, 8.1 Hz, 1H), 8.13 (dd, *J* = 34.3, 8.2 Hz, 1H), 7.80 (dd, *J* = 13.1, 6.9 Hz, 1H), 7.47–7.36 (m, 1H), 7.12 (d, *J* = 4.7 Hz, 1H), 6.92–6.85 (m, 1H), 4.70–4.59 (m, 1H), 4.30–4.18 (m, 1H), 3.91–3.67 (m, 1H), 3.10–3.02 (m, 1H), 2.82–2.69 (m, 2H), 2.45 (ddd, *J* = 14.5, 7.7, 4.1 Hz, 1H), 1.75–1.41 (m, 4H), 0.93 (d, *J* = 6.6 Hz, 3H); ^13^C NMR (151 MHz, DMSO) δ 173.69, 169.78, 158.86 (C-F, q, ^2^*J*_C-F_ = 31.9 Hz), 153.95, 144.62, 143.24, 142.62, 133.65, 133.47, 133.29, 126.19, 124.50, 124.30, 119.64, 117.59, 116.05, (C-F, q, ^1^*J*_C-F_ = 292.8 Hz), 115.69, 113.79, 52.16, 51.82, 50.60, 44.28, 38.96, 36.87, 36.04, 29.27, 28.04, 26.16, 23.89, 23.04, 22.88; MS (ESI) (*m*/*z*): 537.31, [M + H^+^]: 538.30.2-{2-[1-benzyl-2-oxo-6-(2-phenoxyacetamido)-1,2,3,4-tetrahydroquinazolin-4-yl]acetamido}-2-methylpropanamide (**CA4**): white solid; yield 75%; ^1^H NMR (600 MHz, DMSO) δ 10.07 (d, *J* = 112.5 Hz, 1H), 7.95 (d, *J* = 13.3 Hz, 1H), 7.48 (d, *J* = 2.0 Hz, 1H), 7.39–7.26 (m, 5H), 7.23 (dd, *J* = 15.1, 7.5 Hz, 2H), 7.18–7.04 (m, 2H), 7.02–6.68 (m, 6H), 5.05 (d, *J* = 11.9 Hz, 1H), 4.80–4.62 (m, 3H), 4.52–4.45 (m, 1H), 2.53 (d, *J* = 6.6 Hz, 2H), 1.34–1.24 (m, 6H); ^13^C NMR (151 MHz, DMSO) δ 176.74, 169.17, 166.70, 158.24, 154.24, 138.43, 133.85, 132.90, 129.96, 128.94, 127.15, 126.81, 124.02, 121.63, 120.05, 118.36, 115.08, 114.41, 67.45, 56.41, 50.70, 45.09, 25.92, 25.40; MS (ESI) (*m*/*z*): 529.23, [M + H^+^]: 530.25.4-{2-{1-(3-ammoniopropyl)-2-oxo-6-[2-(thiophen-2-yl)acetamido]-1,2,3,4-tetrahydroquinazolin-4-yl}acetamido}-4-carbamoylpiperidin-1-aminium 2,2,2-trifluoroacetate (**CA5**)**·TFA**: yellowish white solid; yield 84%; ^1^H NMR (600 MHz, DMSO) δ 10.18 (s, 1H), 8.57 (d, *J* = 8.9 Hz, 1H), 8.35 (d, *J* = 9.0 Hz, 1H), 8.14 (s, 1H), 7.79 (s, 3H), 7.49–7.42 (m, 2H), 7.39 (dd, *J* = 4.4, 2.0 Hz, 1H), 7.23 (s, 1H), 7.06 (d, *J* = 2.6 Hz, 1H), 7.02–6.97 (m, 4H), 4.69 (td, *J* = 6.6, 2.7 Hz, 1H), 3.94–3.88 (m, 2H), 3.85 (s, 2H), 3.13–2.84 (m, 6H), 2.64–2.54 (m, 2H), 2.13–2.06 (m, 2H), 1.97 (d, *J* = 10.6 Hz, 2H), 1.87 (ddd, *J* = 23.3, 14.5, 7.2 Hz, 2H); ^13^C NMR (151 MHz, DMSO) δ 174.92, 169.92, 168.24, 158.58 (C-F, q, ^2^*J*_C-F_ = 32.3Hz), 154.21, 137.63, 136.59, 133.79, 133.33, 127.15, 126.78, 125.54, 124.00, 119.58, 117.55 (C-F, q, ^1^*J*_C-F_ = 305.48 Hz), 113.91, 55.89, 50.19, 44.39, 39.11, 37.87, 37.17, 29.17, 28.02, 25.61; MS (ESI) (*m*/*z*): 527.23, [M + H^+^]: 528.20.*N*-[4-(2-amino-2-oxoethyl)-1-methyl-2-oxo-1,2,3,4-tetrahydroquinazolin-6-yl]nicotinamide (**CA6-a**): yellow solid; yield 74%; ^1^H NMR (600 MHz, DMSO) δ 10.48 (s, 1H), 9.17 (s, 1H), 8.82 (s, 1H), 8.42 (d, *J* = 7.9 Hz, 1H), 7.68 (dd, *J* = 8.5, 1.8 Hz, 2H), 7.59 (s, 1H), 7.35 (s, 1H), 7.03 (s, 1H), 7.00–6.90 (m, 2H), 4.68 (t, *J* = 5.4 Hz, 1H), 3.19 (s, 3H), 2.44 (dd, *J* = 6.8, 2.1 Hz, 2H); ^13^C NMR (151 MHz, DMSO) δ 171.53, 163.58, 154.49, 151.18, 148.20, 147.99, 137.31, 135.56, 133.25, 124.59, 124.42, 120.77, 118.68, 113.56, 50.20, 43.85, 29.64; MS (ESI) (*m*/*z*): 339.13, [M + H^+^]: 340.15.*N*-[4-(2-amino-2-oxoethyl)-1-methyl-2-oxo-1,2,3,4-tetrahydroquinazolin-6-yl]-2-phenoxyacetamide (**CA6-b**): white solid; yield 78%; ^1^H NMR (600 MHz, DMSO) δ 10.04 (s, 1H), 7.57 (dd, *J* = 8.8, 2.2 Hz, 1H), 7.43 (d, *J* = 2.1 Hz, 1H), 7.34–7.30 (m, 3H), 7.02–6.97 (m, 4H), 6.95–6.90 (m, 2H), 4.68–4.63 (m, 3H), 3.16 (s, 3H), 2.41 (dd, *J* = 6.8, 3.8 Hz, 2H); ^13^C NMR (151 MHz, DMSO) δ 171.50, 166.66, 158.30, 154.47, 135.24, 132.93, 129.98, 124.43, 121.63, 120.10, 118.04, 115.14, 113.55, 67.54, 50.15, 43.77, 29.61; MS (ESI) (*m*/*z*): 368.15, [M + H^+^]: 369.15.2-{6-[(2-methoxyphenyl)sulfonamido]-1-methyl-2-oxo-1,2,3,4-tetrahydroquinazolin-4-yl}acetamide (**CA7-a**): white solid; yield 84%; ^1^H NMR (600 MHz, DMSO) δ 9.71 (s, 1H), 7.68 (dd, *J* = 7.8, 1.6 Hz, 1H), 7.56–7.52 (m, 1H), 7.27 (s, 1H), 7.17 (d, *J* = 8.3 Hz, 1H), 6.99 (t, *J* = 7.6 Hz, 1H), 6.95 (dd, *J* = 8.7, 2.4 Hz, 2H), 6.92 (d, *J* = 2.3 Hz, 1H), 6.84 (s, 1H), 6.78 (d, *J* = 8.8 Hz, 1H), 4.54 (td, *J* = 6.8, 2.9 Hz, 1H), 3.92 (s, 3H), 3.07 (s, 3H), 2.28 (ddd, *J* = 50.0, 14.7, 6.9 Hz, 2H); ^13^C NMR (151 MHz, DMSO) δ 171.24, 156.76, 154.36, 135.83, 135.38, 131.95, 130.74, 126.64, 124.74, 121.05, 120.49, 119.14, 113.80, 113.17, 56.52, 49.87, 43.61, 29.55; MS (ESI) (*m*/*z*): 404.12, [M + H^+^]: 405.10.2-{6-[(2-fluorophenyl)sulfonamido]-3-[(2-fluorophenyl)sulfonyl]-1-methyl-2-oxo-1,2,3,4-tetrahydroquinazolin-4-yl}acetamide (**CA7-b’**): white solid; yield 76%; ^1^H NMR (600 MHz, DMSO) δ 7.93–7.78 (m, 4H), 7.54 (t, *J* = 8.2 Hz, 2H), 7.47 (dt, *J* = 12.2, 7.7 Hz, 2H), 7.35 (s, 1H), 7.20 (d, *J* = 3.0 Hz, 1H), 7.10 (dd, *J* = 8.7, 2.1 Hz, 1H), 7.01 (dd, *J* = 7.6, 5.5 Hz, 2H), 6.94 (s, 1H), 4.65 (td, *J* = 6.8, 3.2 Hz, 1H), 3.18 (s, 3H), 2.32 (ddd, *J* = 21.4, 15.0, 6.9 Hz, 2H); ^13^C NMR (151 MHz, DMSO) δ 171.11, 159.59, 157.89 (C-F, q, ^1^*J*_C-F_ = 254.62 Hz), 153.93, 141.03, 138.46, 132.04, 131.89, 129.12, 125.74, 125.64, 124.99, 118.18, 114.20, 49.48, 43.65, 29.86; MS (ESI) (*m*/*z*): 550.08, [M + H^+^]: 551.10.

### 3.3. Biological Evaluation

#### 3.3.1. In Vitro Anticancer Activity

An MTT assay was used to evaluate the in vitro cytotoxicity of the new compounds. The MTT assay depends on the reduction of soluble 3-(4,5-methyl-2-thiazolyl)-2,5-diphenyl-tetrazolium bromide (MTT) into a blue purple formazan product, mainly by mitochondrial reductase activity inside living cells. The cells used in the cytotoxicity assay were cultured in RPMI 1640 supplemented with 10% fetal calf serum. The cells were suspended in growth medium and plated in 96-well culture plates at a density of 4 × 10^4^ cells/well and incubated at 37 °C in a 5% CO_2_ incubator. After 16 h of incubation, the cells were treated with a series of concentrations (2.5, 5, 10, 20, 40, 80, 160 μM) of each quinazolinone derivative. The negative control received the same concentration of DMSO. After a 72-h incubation, 100 μL of a 0.5 mg/mL MTT solution was added, and the cells were further incubated for 1.5 h at 37 °C. The supernatant was carefully removed from each well and 100 μL of DMSO was added to each well to dissolve the formazan crystals that were formed by the cellular reduction of MTT. After mixing with a mechanical plate mixer, the absorbance of each well was measured in a microplate reader at a wavelength of 570 nm. The results are expressed as the IC_50_ (μM), the concentration that caused 50% inhibition of cell growth of the treated cells when compared to the growth of control cells. Each experiment was performed at least 3 times.

#### 3.3.2. DPPH• Assay for Measurement of Antiradical Activity

The antiradical activity of the compounds was estimated according to a slight modification of the procedure reported by Morales and Jimenez-Perez. [55] Nineteen compounds were diluted in ethanol to l0, 20, 40, 80, 160, and 320 µM. Each sample (100 μL) at a different concentration was added to a freshly prepared solution of 1,1-diphenyl-2-picrylhydrazyl (DPPH•) radical (0.2 mM, 100 μL). The mixtures (200 μL) were placed in a 96-well microplate, shaken vigorously, and then left to stand for 30 min in the dark. Then, absorbance was immediately measured at a wavelength of 517 nm. Vc was used as the standard. The antiradical activity for each compound was determined in the Vc equivalent antioxidant capacity. The DPPH• solution in the presence of ethanol and in the absence of compounds was tested and used as a negative control. A null DPPH free radical scavenging for ethanol was verified. In all experiments, the samples were analyzed in triplicate, and activity for each compound was presented as the mean value ± SEM and used for evaluation of structure–activity relationships. The IC_50_ value represents the concentration of the compound required to scavenge 50% of the DPPH radicals. Absorbance was taken at 520 nm using a microplate reader (Tecan, Männedorf, Switzerland). The percentage of scavenging activity was calculated by the given formula:% scavenging activity = (1 − Abs/Abc) × 100
Abs = sample absorbance; Abc = control absorbance.

### 3.4. Computational Evaluation

#### 3.4.1. Bioinformatics Prediction

Public databases and related software were used to predict the molecular mechanism of quinazoline-2-ones for cancer treatment.

(a)Prediction of disease targets

First, compounds **CA1-e** and **CA1-g** with good anticancer activity based on the MTT assay, were drawn with ChemDraw 14.0.0.117, (CambridgeSoft, Boston, MA, USA), and their SMILE numbers were duplicated. Through the Swiss Target Prediction website (http://www.swisstargetprediction.ch/ (accessed on 6 November 2021)), the SMILE numbers of compounds **CA1-e** and **CA1-g** were entered and the corresponding targets were retrieved. Next, 937 and 3668 disease genes were obtained from NCBI’s GeneCards database (https://www.ncbi.nlm.nih.gov/pmc/ (accessed on 20 November 2021)) for liver cancer and ovarian cancer. After drawing the target compound and disease-related targets as a Venn diagram, using Venny 2.1.0 (https://bioinfogp.cnb.csic.es/tools/venny/ (accessed on 20 November 2021)), 10 overlapping potential quinazolinone targets were identified for the treatment of liver and ovarian cancer.

(b)Protein interaction (PPI) network analysis

To explore the mechanism of quinazoline-2-ones, the protein type was set to Homo sapiens, and the confidence was set at medium (0.400) in the STRING database (https://cn.string-db.org/(accessed on 20 November 2021)). The 10 overlapping targets were input to establish a PPI network. The target network had 10 nodes and 28 edges, and the average node degree was 5.6 (Figure 5a). The PPI network data built in the STRING platform was then imported into Cytoscape_v3.9.1 (National Institute of General Medical Sciences, Bethesda, MD, USA), and the first 10 core targets were selected sequentially using the MCC method in the CytoHubba plug-in (Figure 5b). The highest scores suggested ERBB2, SRC, TNF, and AKT1 may play a vital role in treating liver and ovarian cancer. In our study, ERBB2 ranked first with a score of 48. Therefore, ERBB2 may become a potential therapeutic intervention for liver and ovarian cancer.

(c)GO annotation and KEGG pathway enrichment analysis

Based on the 10 quinazolinone targets obtained with the PPI network, 10 biological processes were screened using GO biological process enrichment analysis. The visual results of enrichment analysis of GO biological processes, using the Cytoscape plug-in ClueGO, are shown in Figure 6a. The most important biological processes were the ERBB2 and IL-6-mediated signaling pathways. The overlapping genes were then introduced into the DAVID database (https://david.ncifcrf.gov/ (accessed on 25 November 2021)). For path enrichment analysis, the recognition type was set to the official gene symbol, the list type was set to GeneList, and the species was limited to Homo sapiens. The significant differences that were identified with KEGG pathway information were indicated by a path bubble diagram drawn by R 3.6.0 (Microsoft, Redmond, USA). The most significant pathways of enrichment were hepatitis B, influenza A, and pulmonary tuberculosis.

(d)Molecular docking

Molecular docking is an essential means of studying possible modes of the action of bioactive compounds. In this study, the crystal structure of the HER2 protein (PDB: 3PP0) was downloaded directly from the PDB database in Discovery Studio software. Then, water and excess conformation were removed and incomplete amino acid residues and hydrogenation were supplemented. The ligand structure was plotted on ChemBioDraw Ultra 14.0 and treated with hydrogenation and energy minimization. Because the protein was a eutectic mixture, the position of the ligand was set as the active center and the sphere’s radius was modified to 10. The CDOCKER RMSD threshold was set to 0.5 to ensure that the conformation of the docking was as diverse as possible, and the docking results were analyzed (Figure 7). The RMSD values were all 0.476 (RMSD is the root mean squared deviation, which compares the structural differences between the docking ligand and the original ligand). The smaller the value, the higher the accuracy of the docking software. An RMSD ≤ 2 is considered an indication that the docking structure is more reliable. The interaction energy is an essential indicator of ligand-receptor affinity, and the greater the negative value, the more stable the binding (Table 4). Each ligand corresponded to 10 different conformations, and the conformation with the lowest docking binding energy was selected as the dominant conformation. In the HER2 binding cavity (Figure 7b), compound **CA1-e** formed hydrogen bonds with SER728, MET801, and ARG849; compound **CA1-g** formed hydrogen bonds with SER728, MET801, CYS805, ASP808, THR862, and ASP863 of HER2 (Figure 7c). In addition, there were van der Waals forces and pi-pi interactions. The amino acid residues and bond length of hydrogen bond interactions are shown in Table 4.

#### 3.4.2. ADMET Studies

Drug properties of the compounds designed according to Lipinski’s five rules are an effective way to select new pharmacologically effective drugs. Statistically speaking, few drugs violate the five rules because the rules govern drug bioavailability. All of our compounds were built and fully minimized using the CHARMM force field through Discovery Studio 2016 software (Accelrys, Inc., San Diego, CA, USA). The low-energy conformers of all of the compounds were used to calculate the ADMET properties (Table 5). The parameters of ADMET were judged according to the levels shown for each parameter. ADMET aqueous solubility levels had six classifications: 0 for extremely low, 1 for no, very low, but possible; 2 for yes, low; 3 for yes, good; 4 for yes, optimal, 5 for no, too soluble, and 6 for molecules with one or more unknown AlogP98 types. Intestinal absorption levels (HIA) were 0, 1, 2, and 3 for good, moderate, low, or very low absorption, respectively. Cytochrome P450 (CYP2D6) inhibition and hepatotoxicity were indicated as 1 for an inhibitor and toxic, or 0 for a non-inhibitor and non-toxic. Plasma protein binding (PPB) was 0 or 1, which showed binding was ≥90%, or <90%, respectively. Blood-brain barrier permeability (BBB) was designated as 0 for very high, 1 for high, 2 for medium, 3 for low, and 4 for undefined.

## 4. Conclusions

In summary, we developed a solid-phase synthesis of dihydroquinazoline-2(1*H*)-ones. By overcoming the shortcomings of existing traditional liquid-phase synthesis methods, such as lengthy steps, tedious steps, and low yield, a simple and efficient solid-phase synthesis method is provided. At the same time, solid phase synthesis technology is a necessary supplement for high-throughput screening of combinatorial chemistry and the feasibility of its synthesis route can shorten the cycle of new drug development. Biological evaluation showed that **CA1-e** and **CA1-g**, with the same substitution at R_1_ and R_2_, were the most effective for HepG-2 and A2780 cells, with IC_50_ values of 37.59 μM, 45.41 μM and 22.76 μM, 22.94 µM, respectively. In addition, anti-free radical DPPH• assay was performed to investigate if the anticancer activity of these quinazoline-2-one compounds could be due to their antioxidant nature. However, the result suggests that compound **CA1-7** is highly capable of DPPH• scavenging with an IC_50_ of **CA1-7** of 57.99 µM, close to that of Vc (22.50 µM). **CA1-e** and **CA1-g**, which are effective at killing cancer cells, showed no antioxidant activity, which suggests that the cytotoxic effect of **CA1-e** and **CA1-g** is not associated with antioxidant properties or related to other factors. Further analysis using bioinformatics prediction was performed to show that ERBB2, SRC, TNF receptor, and AKT1 are potential key targets and play an essential role in cancer treatment. In the subsequent molecular docking, the binding mode of the compound **CA1-e** and **CA1-g** and HER2 (ERBB2) active pocket was shown. In addition, the structure–activity relationship between the small molecular quinazoline-2-one compounds and their anticancer activity developed in this study provides a theoretical basis and concrete evidence for the development of small molecule anticancer drugs.

## Data Availability

The data presented in this study are available in the Appendix A.

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
