# Peer review of "Novel Approaches for the Solid-Phase Synthesis of Dihydroquinazoline-2(1H)-One Derivatives and Biological Evaluation as Potential Anticancer Agents"

_molecules, 2022, doi:10.3390/molecules27238577_

Round 1

Reviewer 1 Report

Wang, Zheng and collaborators furnish in this paper a solid-phase approach to biologically active dihydro-2 quinazoline-2(1H)-one derivatives. Efficient synthetic strategy gains the authors a well-varied chemical library, where the synthetic scope is extended to the three substituents on the 1,4-disubstituted 3,4-dihydro-2 (1H)-quinazolinones scaffold. Evaluation of the anticancer activities evidenced a pronounced effect of the 4-substituent on the aromatic moiety on the overall cytotoxicity, although no compound shows significant anticancer activity, when compared to the reference literature. The experimental study is corroborated with a detailed bioinformatics prediction study. The experimental section is well-crafted and articulated. We advise publishing the article on Molecules with the minor changes listed below.

A literature reference for anticancer activity for the selected cell lines should be added in Table 2.

Line 177. Change “An” to “A”.

Structure numbers should be in bold throughout the text.

Line 251. Change “1HNMR” to “1H NMR”.

Figure 2. a) Fmoc structure misses a methylene unit; b) Spectrometer working frequency and solvent of choice should be reported.

Line 317. Change “electron-attracting” to “electron-withdrawing”.

Reviewer 2 Report

In the current manuscript entitled “Novel Approaches for the Solid-Phase Synthesis of Dihydro-2 quinazoline-2(1H)-one Derivatives and Biological Evaluation as 3 Potential Anticancer Agents” authors developed new series of small molecules as potential anticancer and antioxidants agents. Before accepting this manuscript for publication in Molecules, the authors should address the following points.

Authors should use positive control and reference drug in both anticancer MTT and antioxidant DPPH• assays.

Authors should evaluate the cytotoxicity of target compounds should be assessed towards normal cell lines.

The cytotoxicity results should be expressed in μM not (μmol/L). Please unify throughout the text.

Authors should consider the obtained cytotoxicity results as week activity, and should justify that in the text.  

Authors stated in the abstract “However, these inhibitors 15 are limited by low oral bioavailability, poor targeting selectivity, and tumor acquisition of mutations 16 leading to drug resistance”. However, authors have not addressed any of these issues! So, I recommend removing this sentence.

The Introduction section contains a lot of non-relevant information, specially the talk about the kinase inhibitors. Please revise the introduction to include only the pertinent information needed for the study's context and background.

The Results and Discussion section for “Chemistry” is too lengthy and should be reduced and be more concise.

It is better if authors provided experimental validation for the predicted kinases targets.

Reviewer 3 Report

The authors developed two novel methods of combinatorial chemistry and successfully synthesized dihydroquinazoline-2(1H)-one derivative. The authors have also done in cell, in vitro, and in silico analyses to clear molecular targets of the related compounds. The authors concluded the compounds are useful for cancer treatment. However, to receive the conclusion, the authors should improve the manuscript.

Major point

1.      The authors need to follow the method of scientific writing, as follows.

1-1. The computational studies are not “biological results”. The authors should create section “2.3 Computational evaluation”

1-2. The conclusion section should be cleaned up. Discussive remarks should write in discussion section.

1-3.  Most part of Lines 349 to 364 (Section 2.2.2) should be moved to introduction section.

1-4.  The Discussion section that connects each result – biological to computational – should be placed after results. For example, placing “2.4 Discussion”, or separate “2. Results and discussion” to “2. Results” and “3. Discussion”. See also Major point 2 and 3.

2.      The author should discuss why CA1-e and g (effective anti-cancer compounds) had no anti-oxidative effect in section 2.2.2., or 2.4. This point is important because it shows the effect is NOT caused by anti-oxidant but caused by targeting unknown target(s) (the authors predicted it as HER2).

3.      It was very interesting results that IC50 values of the compounds differ in cell lines. Especially, comparing breast cancer cells (MDA-MB-231) and ovarian cancer cells (A2780), the effect pattern of CA1e,f, and g were different. The authors should discuss about this point. For example, MDA-MB-231 has HER2 amplification, according to ECACC catalog (https://www.culturecollections.org.uk/media/133182/mda-mb-231-cell-line-profile.pdf ), on the other hand, the endogenous HER2 expression in A2780 is low (https://www.ncbi.nlm.nih.gov/pmc/articles/PMC6422889/ ). If the authors predict the target as HER2, the authors should discuss this point.

4.      The control compounds that have quinazolinones should be asked for anti-proliferative activity, such as DQO-1, PFI-1, or others.

5.      In docking study, control compound is needed, such as TAK-285. It is recommended to use crystal structure of an inhibitor-receptor complex, for example, 3RCD.

6.      3PP0 is a crystal structure of HER2 dimer. In the docking results shown in Figure 7, CA1-e and CA1-h were docked with different site corresponding position of each subunit. Please limit the binding pocket to one subunit. The illustration of amino acids in active pocket should be aligned to clear the difference (or same point) between CA1-e, CA1-h, and control compounds.

Minor point

1.      The term “in vitro” should be italic (please check all).

2.      The resolution of Figure 5 and 6 are little low.

Round 2

Reviewer 2 Report

The manuscript has been improved, and should be accepted now. 

Author Response

Dear Reviewer,

Thank you for your comments concerning our manuscript entitled “Novel Approaches for the Solid-Phase Synthesis of Dihydroquinazoline-2(1H)-one Derivatives and Biological Evaluation as Potential Anticancer Agents” (ID: molecules-2042211). Those comments are all valuable and very helpful for revising and improving our paper, as well as the important guiding significance to our work.

Reviewer 3 Report

The authors did great improve of the manuscript in the first round. I apologize for the mistaking that I wrote MDA-MB-231 has HER2 amplification. However, the authors correct the mistake and rewrite the manuscript appropriately.

However, the core point of my premier comment (point 3) has not referred in the manuscript in round 2. In the point of the structure-activity relationship study, the authors should compare compounds which have activity and not have activity as the author wrote in 2.2.1. For example, in your manuscript, CA1-e (22.76 μM: A2780) and CA1-f(> 100 μM: A2780) should be compared.

The authors consider the two pyridine-containing compounds (e and f) as greater cell proliferation inhibitory activity (Lines 243-245), but this ignores the result of A2780. The authors refer this comparison in 2.2.1 or 2.3.1.

Author Response

Point 1: However, the core point of my premier comment (point 3) has not referred in the manuscript in round 2. In the point of the structure-activity relationship study, the authors should compare compounds which have activity and not have activity as the author wrote in 2.2.1. For example, in your manuscript, CA1-e (22.76 μM: A2780) and CA1-f (> 100 μM: A2780) should be compared.

The authors consider the two pyridine-containing compounds (e and f) as greater cell proliferation inhibitory activity (Lines 243-245), but this ignores the result of A2780. The authors refer this comparison in 2.2.1 or 2.3.1.

Response 1: We are so sorry that we did not fully reply to your question in the manuscript in round 2. Your suggestion is indeed very thoughtful and helpful. As suggested by the referee, we corrected the previous misdescription in the structure-activity relationship study in section 2.2.1 and discussed it in 2.3.1.